# An Optical and Chemiluminescence Assay for Assessing the Cytotoxicity of *Balamuthia mandrillaris* against Human Neurospheroids

**DOI:** 10.3390/bioengineering9070330

**Published:** 2022-07-20

**Authors:** Worakamol Pengsart, Kasem Kulkeaw

**Affiliations:** 1Graduate Study School, Faculty of Medicine Siriraj Hospital, Mahidol University, Nakhonpathom 73170, Thailand; worakamol.pen@student.mahidol.edu; 2Siriraj Integrative Center for Neglected Parasitic Diseases, Department of Parasitology, Faculty of Medicine Siriraj Hospital, Mahidol University, Bangkok 10700, Thailand

**Keywords:** granulomatous amoebic encephalitis, *Balamuthia mandrillaris*, neurospheroid, cytotoxicity

## Abstract

A spheroid is a cell aggregate in a three-dimensional context; thereby, it recapitulates the cellular architecture in human tissue. However, the utility of spheroids as an assay for host–parasite interactions remains unexplored. This study demonstrates the potential use of neurospheroids for assessing the cytotoxicity of the life-threatening pathogenic amoeba *Balamuthia mandrillaris*. The neuroblastoma SH-SY5Y cells formed a spheroid in a hanging drop of culture medium. Cellular damage caused by *B. mandrillaris* trophozoites on human neuronal spheroids was observed using microscopic imaging and ATP detection. *B. mandrillaris* trophozoites rapidly caused a decrease in ATP production in the spheroid, leading to loss of neurospheroid integrity. Moreover, 3D confocal microscopy imaging revealed interactions between the trophozoites and SH-SY5Y neuronal cells in the outer layer of the neurospheroid. In conclusion, the neurospheroid allows the assessment of host cell damage in a simple and quantitative manner.

## 1. Introduction

Given their simple preparation and low-cost maintenance in a standard laboratory, human cancer cell lines or immortalized cells are conventionally cultured and widely used for the study of disease mechanisms and the development of treatments. Typically, these cells grow as a layer on a plastic surface with or without coating with matrix protein. Thus, cells interact with surrounding cells and the surface in a two-dimensional (2D) manner. Nevertheless, advances in biomaterials and bioengineering have led to the establishment of a 3D-culture platform, in which cells proliferate and aggregate in a scaffold or semisolid medium without attaching to a plastic surface [1,2]. This way, cells can behave like human tissue, including cell-to-cell interactions in all directions, gradient exposure to nutrients and gases, cell polarization, and cell signaling pathways. Studies have reported the physiological relevance of 3D culture regarding cell functions [3], drug response [4,5], and cell-mediated cytotoxicity [6]. Prospectively, preclinical platforms for modeling human diseases and drug discovery have been changed from the 2D to 3D platform [7,8].

An amoeba, *Balamuthia mandrillaris*, generally lives in soil and fresh water [9]. However, *B. mandrillaris* is life-threatening when the trophozoite enters the human brain parenchyma. Together with the immune response, the brain-invading *B. mandrillaris* causes neuroinflammation called granulomatous amoebic encephalitis (GAE) in both immunocompromised and immunocompetent hosts [10]. GAE is extremely rare, but it is highly lethal. A few GAE patients survived among the more than 200 cases reported worldwide. Most patients died due to the lack of a sensitive diagnosis or a specific treatment regimen [11]. None of the current drug regimens are radical cures, and all regimens vary and heavily rely on combinations of antimicrobial and antifungal drugs [12,13]. Thus, effective drugs are urgently needed. The development of a treatment primarily uses a culture of *B. mandrillaris* trophozoite in the pathogenic stage placed in a human cell-free well of a cell culture plate. A killing dose of a potent compound is an indicator of amoebicidal activity [14,15]. However, this technique entirely excludes host–parasite interactions, which are a crucial part of disease progression and severity. Thus, cytotoxicity-based assays would better allow for the discovery of drugs that ameliorate disease severity.

The human neuroblastoma SH-SY5Y cell line has been used for the study of Parkinson’s disease [16] and other neurodegenerative disorders, such as Alzheimer’s disease [17]. Compared to the human brain, the in vitro 2D culture of neuroblastoma SH-SY5Y cells failed to recapitulate neurodegenerative disorders because of differences in tissue organization and cell-to-cell and cell–extracellular matrix interactions [18]. Three-dimensional organizing cells have been deployed as disease models to accurately mimic human tissues [19]. There are many 3D-based platforms established as disease models and drug screenings. Spheroids are 3D-organizing cells generated using a physical force to aggregate cancer cells in a spherical shape or allow cancer cells to proliferate by embedding them in a semisolid medium. Therefore, the tumor spheroid has a cell architecture similar to that of tumors in the human body.

This study aimed to demonstrate tumor spheroids as an assay to assess the cytotoxicity of pathogenic amoebae. To mimic the 3D architecture of the brain parenchyma, neurospheroids were generated by culturing human neuroblastoma SH-SY5Y cells in a hanging drop of cell suspension. The neurospheroids were cocultured with clinically isolated *B. mandrillaris* trophozoites at the pathogenic stage. Cytotoxicity was monitored using a fluorescence-based and a label-free microscope. Collectively, the use of neurospheroids allows the assessment of the cytotoxic effects of *B. mandrillaris*. Thus, the neurospheroid poses an in vitro platform for investigating therapeutic targets in the context of the host–parasite interaction under a more physiologically relevant setting relative to 2D culture.

## 2. Materials and Methods

### 2.1. Culture of B. mandrillaris from the Biopsied Brain of a Human Subject

The *B. mandrillaris* trophozoites were clinically isolated from a human subject with consent to participate (COA no. Si806/2020). Briefly, the biopsied brain was chopped into small pieces and subjected to pepsin digestion. The large debris was removed by passing the pepsin-digested brain tissue through multiple layers of sterile gauze bandage. After three washes with penicillin/streptomycin-containing PBS, the cell pellet was resuspended in 10% DMEM supplemented with 10% heat-inactivated fetal bovine serum (FBS, HyClone, Utah, USA) and co-cultured with human lung carcinoma A549 cells. The culture medium was changed every 2–3 days, and the feeder cells were renewed every week. After complete removal of the A549 feeder cells, the trophozoites were subsequently transferred to the culture of human neuroblastoma SH-SY5Y cells.

### 2.2. Culture of Human Neuroblastoma SH-SY5Y Cells

Human neuroblastoma SH-SY5Y cells obtained from the American Type Culture Collection (ATCC^®^ No. CRL-2266TM) were routinely cultured following the ATCC’s instructions. A mixture of ATCC-formulated Eagle’s minimum essential medium (EMEM) (ATCC, USA) and F12 medium (Gibco, Gaithersburg, MD) was prepared at a 1:1 ratio (hereafter called EMEM-F12). To prepare a complete medium for human SH-SY5Y cells, EMEM-F12 medium was supplemented with 10% heat-inactivated fetal bovine serum (FBS, HyClone, USA), herein called complete EMEM-F12. Cells were incubated at 37 °C in a humidified atmosphere containing 5% CO_2_. At a cell density of 60–80% confluence, cells were subcultured using 0.25% trypsin in 0.5 mM EDTA solution (STEMCELL Technologies, Vancouver, Canada). The number of viable cells was counted using Trypan blue and a hemocytometer.

### 2.3. Formation of Human Neurospheroids

A total of 2 × 10^4^ viable SH-SY5Y cells were prepared in a volume of 30 µL of complete EMEM-F12 and added to a 96-well hanging drop plate (SPL Life Sciences, Gyeonggi, South Korea). Then, PBS solution was added to the reservoir to prevent evaporation of hanging drops. After 1–2 days of incubation, spheroids were transferred to a cell floater plate (a round bottom, low-attachment 96-well culture plate; SPL Life Sciences, South Korea) by pipetting 70 µL of the complete EF12 on top of the well.

### 2.4. Culture of B. mandrillaris and Cocultures with Human Neurospheroids

Regular culture of the clinical isolate of *B. mandrillaris* was performed following a previous study [20]. The trophozoites were subcultured when the human SH-SY5Y cells were destroyed by more than 80%. The trophozoites were plated onto 80–90% confluent SH-SY5Y cells. For coculture, 200 amoeba/20 µL of the trophozoites were cultured with SH-SY5Y neurospheroids in complete EF12 medium. Cells were incubated at 37 °C in a humidified atmosphere containing 5% CO_2_. Cells were monitored using an inverted microscope with 20× magnification. Due to an imperfect spherical shape, a total of 10 axes lining through the center and ending at the perimeter of the spheroid were set and measured using ImageJ software. The lengths of the axes were calculated for a mean, and displayed as the diameter of the neurospheroids.

### 2.5. Cell Labeling with Fluorescent Dye and Confocal Microscopy

To examine cell-to-cell interactions, human neuroblastoma SH-SY5Y cells were labeled with protein-binding fluorescent dyes in a 2D culture, followed by spheroid formation. Briefly, the monolayered SH-SY5Y cells were washed with PBS. A 2.5 µM CellTracker™ Green CMFDA (Invitrogen, Oregon, USA) was prepared in complete EF12 medium and incubated with the cells at 37 °C in a humidified atmosphere containing 5% CO_2_ for 45 min. Then, the CMFDA-labeled cells were harvested using trypsin solution and subjected to spheroid formation. To label lipids in the trophozoites, the *B. mandrillaris* trophozoites were incubated with a 1:1600-diluted Vybrant™ DiD Cell-Labeling Solution (Invitrogen, Oregon, USA) for 15 min. After washing with PBS, the CMFDA-labeled neurospheroid was cultured with DiD-labeled *B. mandrillaris* trophozoites within 3 h post-cell labeling. To monitor cell-to-cell interactions in 2D and 3D modes, the cultures were seeded in a low attachment 96-well plate. Microscopic images were captured using confocal microscopy (Nikon, Tokyo, Japan).

### 2.6. Reverse-Transcription PCR

To examine the differentiation and survival status of SH-SY5Y neuroblastoma cells, the transcript levels were examined using reverse-transcription PCR. RNA was extracted from the cells, and DNA was eliminated using an RNA extraction kit following the manufacturer’s instructions (Favorgen, Pingtung, Taiwan). The cDNA of the extracted mRNA was synthesized using the iScript reverse transcription supermix. cDNA was amplified using reverse-transcription PCR (Bio-Rad, California, USA). An RNA sample, which was not reverse transcribed, was used as the no-RT control. Primer sets were used following previously published articles and are described in Table 1. Relative expression of the transcript was compared among samples by normalizing with *ACTB* mRNA and calculated using the delta cycle threshold.

### 2.7. Cell Viability Assay

The CellTiter-Glo^®^ 3D Cell Viability Assay was used to quantify ATP, an indicator of metabolically active cells. The chemiluminescent signal of ATP was measured at 490 nm and displayed as relative light units (RLUs).

### 2.8. Statistical Analysis

The Mann–Whitney test, which compares the difference between two independent samples, was used to compare the mean spheroid size. The student’s *t*-test was used for examining the difference in means of relative expression of genes and the RLUs of ATP levels. A *p* value less than 0.05 was set as statistically significant.

## 3. Results

### 3.1. Microscopic Observation of Human Neurospheroids

To minimize the physical barrier between *B. mandrillaris* trophozoites and spheroids, a scaffold-free method was employed. A suspension of human SH-SY5Y cells was hung as a drop at a 9 mm depth (Figure 1A). Given the difficulty in capturing images of cells in the drops by using an inverted microscope, a stereomicroscope was employed despite the poor resolution of the captured images (Figure 1A). After 6 h of hanging, cells clumped at the center, and some cells scattered around the cell clump (Figure 1A, below the schematic diagram). The cell clump was translucent. After 24 h of hanging, the cell aggregates became more compact, as indicated by less translucence, but retained an irregular shape. To observe neurospheroids, the cell aggregates in the hanging drop were transferred to a low-attachment U-shaped well and observed using an inverted microscope. Cell aggregates became spherical and dense with a well-defined edge at 48 h post-culture. A small area of dark zone was observed at the center, resembling a hypoxic core where low oxygen causes cell death [25,26] (Figure 1B). At 72 h, a larger area of dark zone was observed at the core of the neurospheroid. The sizes of the 48- and 72-h neurospheroids were slightly different: 500 μm × 300 μm and 700 μm × 250 μm, respectively. In contrast, the day-3 neurospheroids possessed a dark zone at the core.

### 3.2. Neuron-Specific Transcript of Human Neurospheroid

To examine the effect of the 3D organization of cells on human SH-SY5Y cells, the maturity of the neurospheroid was compared with that of the 2D culture regarding the transcription of neuronal genes. The levels of transcripts encoding filament Nestin (NES), a neuroblast marker, were not different among the conventional cell culture and the neurospheroid. By contrast, the neurospheroid expressed transcripts of immature neuron, tubulin beta 3 class III (TUBB3 or TUJ1) and neuronal differentiation 1 (NEUROD1), at a level higher than the 2D culture did. Despite a similar level of microtubule associated protein 2 (MAP2), the neurospheroid expressed transcripts of synaptophysin (SYP), the maturation marker of neurons, more highly than that of the 2D culture counterpart at days 2 and 3 of hanging (upper panel, Figure 1C). Thus, the 3D neurospheroid upregulated the expression of neuron-specific genes.

### 3.3. Decrease in the Size of Neurospheroids Post Coculture with B. mandrillaris Trophozoites

Coculture of human neurospheroids with *B. mandrillaris* trophozoites was performed in a low-attachment U-shaped well. Owing to the roundness of the U-shaped well, the trophozoites were proximal to the neurospheroid by gravity (lower panel of Figure 2A, arrowheads). After 24 h post-coculture with the trophozoites, the edge of the neurospheroid was ill-defined (Figure 2B). A higher magnified view of Figure 2A shows the cytoplasm-protruding trophozoites attached at the edge. At 48 and 72 h post-coculture, there was cellular debris and cells surrounding the neurospheroid. The non-translucent area was decreased. At 72 h of coculture, there was only a dark core of the neurospheroid remaining. Next, the cytotoxic effect of *B. mandrillaris* trophozoites was assessed by measuring the size of the neurospheroid. The size of the untreated spheroids remained unchanged together with the expansion of the dark zone (Figure 2A,C). In the presence of *B. mandrillaris* trophozoites, the size of the neurospheroid significantly decreased from day 1 onward (Figure 2C).

### 3.4. Cytotoxicity Assay

To deploy the 3D neurospheroid as an in vitro assay, the production of ATP was examined and used as a surrogate marker of cell viability. Given an increase in the number of trophozoites during the period of coculture, the ATP-producing trophozoites confounded the measurement of the neurospheroid-derived ATP. Thus, the level of ATP was normalized to the trophozoite culture. The chemiluminescence signal of the neurospheroid was subtracted from the signal of an equal number of *B. mandrillaris* trophozoites. The cytotoxic effect was calculated as the percentage of nonviable cells in a neurospheroid. At 24 and 48 h post-culture, there was no significant difference in the level of ATP (Figure 3A). However, when comparing the value of chemiluminescence with the starting point (0 h), there was a tendency toward a decrease in ATP levels, implying a negative effect on human cell metabolism (Figure 3B). Without the neurospheroids, the *B. mandrillaris* trophozoites decreased ATP production about 2-fold post the 24- and 48-h cultures, suggesting a downregulation of metabolism. Moreover, the use of 200 nM Wortmannin for inhibiting trogocytosis failed to abolish the negative effect of the *B. mandrillaris* trophozoites (Figure 3C). Thus, the *B. mandrillaris* trophozoites likely altered the metabolism of the neurospheroid.

### 3.5. Properties of the Neurospheroid at the Molecular Level

To assess the effect of *B. mandrillaris* trophozoites on the regulation of gene expression, the levels of neuron-specific and apoptosis-regulating transcripts were examined. Downregulation of the *NEUROD1* gene was observed at 48 h post-coculture. In contrast, the level of the proapoptotic *BAX* transcript was slightly decreased in the *B. mandrillaris*-cultured neurospheroid (Figure 3D). These results together suggest that *B. mandrillaris* likely affected the expression of cell survival-regulated genes.

### 3.6. 3D Imaging of Neurospheroids

To investigate the ingestion of the cytoplasmic protein of neurospheroids, trophozoites and human neurospheroids were labeled with DiD and CMFDA, respectively, to distinguish human cells under a confocal microscope. At 3 h coculture, the neurospheroid was invaded by the trophozoites (Figure 4A). The morphology of the invading trophozoites lacked cytoplasm projection (arrowheads, Figure 4B), while the trophozoites distal from the spheroid had protruding cytoplasm (arrows, Figure 4B). A zoomed-in image revealed penetration of the trophozoites into a layer of the neurospheroid (arrowheads, Figure 4C).

According to a time-lapse video obtained using the confocal microscope, the DiD-labeled trophozoites attached to the edge of the neurospheroid and transformed to a round shape within 4 min (Figure 5A and Appendix A). The 3D image of the neurospheroid shows that the trophozoites localized densely at the edge (Figure 5B and Appendix A). Higher magnification of a cross-section view revealed an anchoring and penetrating pattern of the *B. mandrillaris* trophozoites (Figure 5D and Appendix A). Moreover, human cytoplasmic protein was surrounded by magenta, implying that the human protein was located in the cytoplasm of the *B. mandrillaris* trophozoites. The 3D view of confocal images revealed the cellular process of *B. mandrillaris* trophozoites interacting with the neurospheroid in an attaching-transforming-invading manner.

## 4. Discussion

The 3D cell culture platform has emerged as a tool for studying the mechanism of human disease and assessing the therapeutic effect of potential small molecules. This study demonstrated a way to deploy neurospheroids for assessing the cytotoxicity of a highly pathogenic, brain-damaging amoeba. Despite the physiological relevancy, the use of a 3D model is hampered by a difficulty in downstream analysis, i.e., morphological observation under a microscope and assessment of cytotoxicity. Thus, the key readouts in this study were the size of the neurospheroid, which primarily relied on optical images of a common inverted microscope without fluorescence. Moreover, in the context of host–parasite interactions, the measurement of intracellular ATP in human neurospheroids was heavily confounded by the ATP of metabolically active trophozoites. Alternatively, a way to calculate the neurospheroid-specific ATP is proposed here.

Spheroids recapitulate the microenvironment of cells in a given tissue in terms of cell-to-cell interactions [7,8]. It could partially imitate the microenvironment in brain tissue, where cells arrange in a 3D manner. Nevertheless, studies demonstrating the use of spheroids as a means to assess the cytotoxicity of pathogens remain limited. Therefore, this study aimed to exemplify the concept of an infectious disease model in a 3D culture. There are several ways to allow cells to form in a 3D manner, including semisolid media or scaffolds. However, these methods need more time to allow penetration of trophozoites through solid media or scaffolds. Thus, a hanging drop was selected to minimize the physical barrier between *B. mandrillaris* trophozoites and spheroids. Moreover, cell analysis in 3D spheroids requires a complicated process, such as cross-sectioning for immunofluorescence. However, a difficult-to-penetrate biochemical assay limits the use of 3D spheroids for the measurement of cytotoxicity. Here, *B. mandrillaris*-mediated cytotoxicity was assessed by measuring the size of spheroids. Despite observation of an obvious decrease in size, the direct measurement was inaccurate due to the proximal contact of dead cells and trophozoites. Moreover, the removal of trophozoites and dead SH-SY5Y cells from spheroids allows more accurate measurement of spheroid size, a situation similar to the cytotoxicity assay of CAR T cells [6]. Ultimately, although there is a simple and low-cost way to assess cytotoxicity, improvement of this proposed culture platform is inevitable.

An optimal way to recapitulate the human brain is to grow and maintain neurons in vitro. However, the isolation and culture of neurons from the human midbrain are hampered by ethical concerns, a shortage of tissue, a limited number of cells, and difficulty in cell expansion. In contrast to the culture of primary cells, the cancer cell line is a renewable, expandable source, overcoming these limitations. Human neuroblastoma is brain cancer generated from immature neurons. The human neuroblastoma SH-SY5Y cell line, a subclone of SK-N-SH cells, was derived from bone marrow-metastasizing neuroblastoma cells, and expresses immature neuronal markers [27,28]. Despite the cancerous origin, the SH-SY5Y cell line has intact genes. Therefore, the SH-SY5Y cell line has been used for the study of neurotoxicity. Regarding neuron gene transcription, the SH-SY5Y-based 3D spheroids displayed a more mature neuron phenotype than the 2D culture without induction by retinoic acid. Collectively, human SH-SY5Y neuroblastoma cells were selected for modeling the cytotoxic effects of pathogenic amoeba.

In the brain parenchyma, *B. mandrillaris* trophozoites must acquire nutrients from the surrounding neurons, leading to host cell damage. Together with attempts to restrict and remove the invading amoeba, host immune cells mount an inflammatory attack that collaterally damages the brain. An assay directly assessing neuronal damage allows findings of non-amebicide targets to inhibit host cell damage. Thus, the mechanism underlying host cell damage is crucial for identifying a cell contact-dependent process in which the parasite must survive. Regarding 3D imaging, our previous work demonstrated the cellular process of host cell ingestion in a 2D culture of SH-SY5Y cells [20]. Given the difference in complexity of microarchitecture, the current study could not observe endocytosis of host cell protein by *B. mandrillaris* trophozoites in coculture with a 3D neurospheroid. With a limitation in live imaging of spheroids, the trophozoites first attached to the outer layer and then transformed into a round morphology lacking cytoplasmic projection. The 3D confocal images reveal the anchoring process, in which the cytoplasm of *B. mandrillaris* trophozoites protruded into a layer of neurospheroid, while some cells were exposed to the culture medium. Moreover, some trophozoites completely submerged into the neurospheroid. Hence, searching for cell contact and invading factors may allow us to inhibit parasite survival. Given that several case studies reported the necrotic lesions in the autopsied tissues of human brain parenchyma [29,30,31,32], suggesting a loss of neuron cells, therefore, the disappearance of the cells in the spheroid was used as an end-point indicator of brain damage.

Nevertheless, the use of human neuroblastoma cell-based spheroid remains limited. Although human cancer cell lines have been used for study of cellular processes, including host cell entry [33], parasite development and cellular alterations [34], the biology of cancer cells is different from that of cells in the body. The alteration of genes in other pathways of SH-SY5Y cells may limit the study of host responses to parasites. Moreover, given the cancer origin of the SH-SY5Y neurospheroid, it is necessary to further confirm the histological and physiological relevancy to a human brain [23]. Hence, a gap in translating the in vitro cancer cell-based models can be narrowed down using a way to recapitulate cell functions in human tissue, including the use of stem cell-derived cerebral organoids for virus infection [35,36] or brain-on-a-chip for fungal brain infection [37].

## 5. Conclusions

The pathogenic *B. mandrillaris* trophozoites cause damage to human neurospheroids, including loss of 3D integrity and a decrease in ATP. Thus, the use of human neurospheroids allows the assessment of host cell damage in a simple and quantitative manner.

## Figures and Tables

**Figure 1 bioengineering-09-00330-f001:**
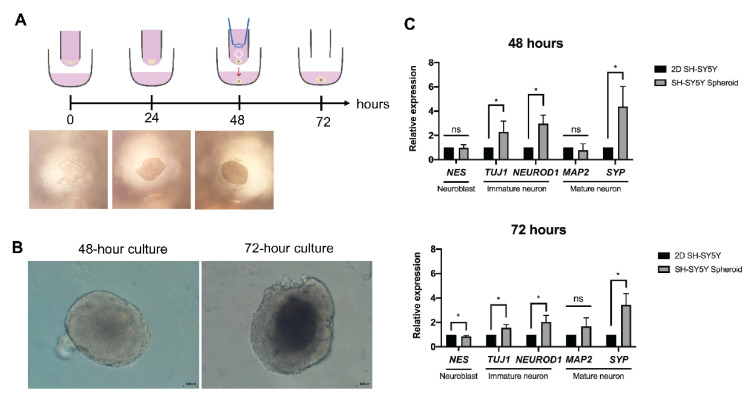
Neurospheroid formation in a hanging drop culture. (**A**) Schematic diagram of spheroid formation in a hanging drop of the culture medium. Images of cell aggregates were visualized using a stereomicroscope. Microscopic images were captured using 4× magnification on a stereomicroscope. (**B**) Spherical shape of the cell aggregate observed under an inverted microscope. Scale bar, 100 μm. (**C**) Relative levels of neuron-specific transcripts in the 3D SH-SY5Y neurospheroid and the 2D cultured SH-SY5Y cells. The data are representative of independent experiments (*n* = 3). Quantitative PCR was performed in a technical triplicate. The statistical test: * *p* < 0.05, ns: not significant.

**Figure 2 bioengineering-09-00330-f002:**
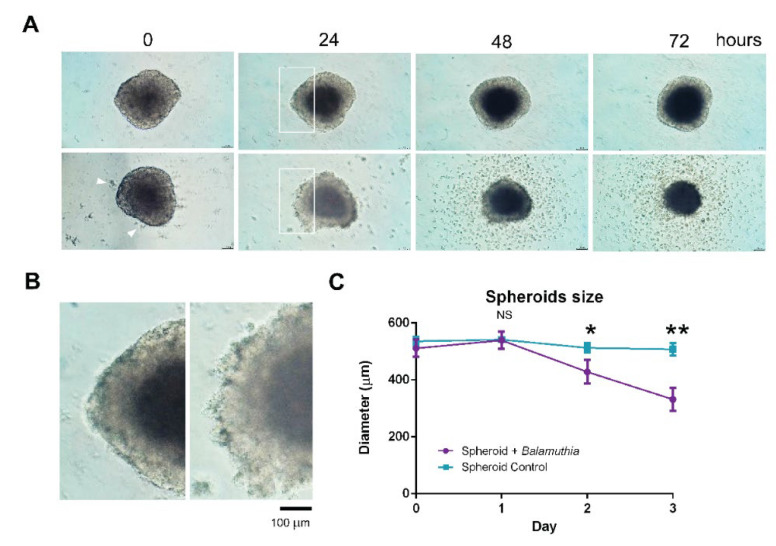
Coculture of neurospheroid with *B. mandrillaris* trophozoites. (**A**) Optical phase contrast images of SH-SY5Y neurospheroids without or with *B. mandrillaris* trophozoites (upper and lower panels, respectively). Scale bar = 200 μm. (**B**) Magnified insets of Figure 2A. Left panel = neurospheroid without *B. mandrillaris* trophozoites (control); Right panel = neurospheroid with *B. mandrillaris* trophozoites (test). (**C**) Diameter of the neurospheroid at different periods of the coculture. Data are expressed as the mean ± standard deviation of the mean (SD) calculated from six spheroids (*n* = 6) for the control and the test. The statistical test: * *p* < 0.01, ** *p* < 0.001; NS: not significant.

**Figure 3 bioengineering-09-00330-f003:**
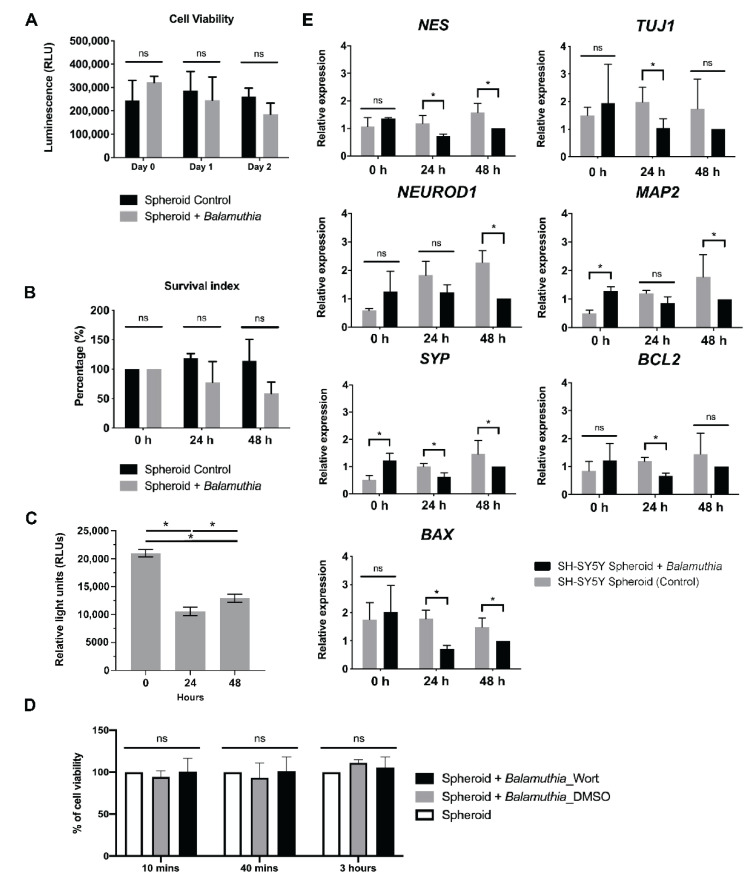
The effects of *B. mandrillaris* trophozoites on the neurospheroid. (**A**) Cell viability of the neurospheroid indicated by ATP level. (**B**) Survival index of the neurospheroid. The survival index was calculated based on the level of ATP at a given time point relative to that of the starting point (0 h). (**C**) The level of ATP produced by *B. mandrillaris* trophozoites in the feeder-free cultures. (D) Inhibition of trogocytosis using 20 nM Wortmannin, an inhibitor of phosphoinositide 3-kinases. (E) Semiquantitative PCR of neuron-specific and apoptosis-related genes. Bar graphs display data of the mean ± standard deviation calculated from technical triplicates. The data are representative of independent experiments (*n* = 3). The statistical test: * *p* < 0.01; ns: not significant.

**Figure 4 bioengineering-09-00330-f004:**
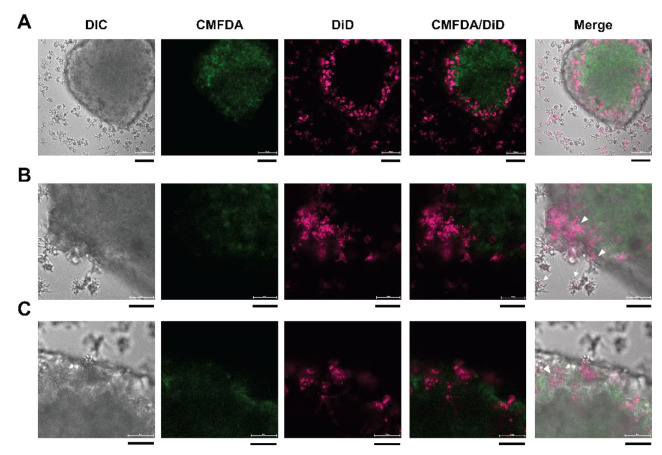
Fluorescence images of the *B. mandrillaris* trophozoites cocultured with human neurospheroid. (**A**) A top view of the neurospheroid. Scale bar = 100 μm. (**B**) Higher magnification at the edge of the neurospheroid. Scale bar = 50 μm. (**C**) Invasion of the trophozoites into a layer of the neurospheroid. Scale bar = 50 μm. DIC, differential interference contrast; CMFDA, 5-chloromethylfluorescein diacetate; DiD, 1,1′-dioctadecyl-3,3,3′,3′-tetramethylindodicarbocyanine, 4-chlorobenzenesulfonate salt.

**Figure 5 bioengineering-09-00330-f005:**
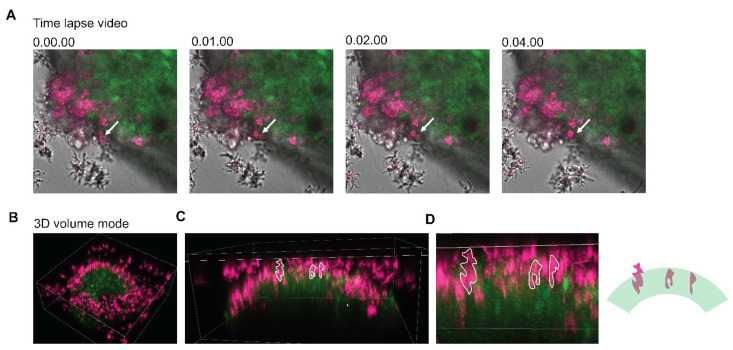
Time-lapse video and 3D view of the neurospheroid cocultured with *B. mandrillaris* trophozoites. (**A**) Host–parasite interaction recorded using the time-lapse mode of a confocal microscope. *B. mandrillaris* trophozoites are magenta, while the human neurospheroid is green. (**B**) The 3D view of the neurospheroid (green) surrounded by the *B. mandrillaris* trophozoites (magenta). (**C**) Cross-section view of the 3D image at the edge of the neurospheroid. Dotted lines are trophozoites (magenta). (**D**) Magnified image of Figure 5C. Trophozoites are marked by a dotted line. In the right panel, the diagram shows the anchoring and penetrating behavior of the *B. mandrillaris* trophozoites (magenta).

**Table 1 bioengineering-09-00330-t001:** List of primer sets for assessment of cell maturation, proliferation and survival.

Target Gene	Primer Sequence (5′ to 3′)	Reference
*Nestin* (Neural progenitor marker)	Fw: CTGTGAGTGTCAGTGTCCCC Rv: CTCTAGAGGGCCAGGGACTT	[21] Almeida et al., 2016
*Tuj1* (Neuron-specific class III beta-tubulin; Early stage neural differentiation marker)	Fw: GCAAGGTGCGTGAGGAGTAT Rv: GTCTGACACCTTGGGTGAGG	[21] Almeida et al., 2016
*NeuroD1* (Mature neuronal marker)	Fw: CCCTTCCTTTGATGGACCCC Rv: AAATGGTGAAACTGGCGTGC	[22] Piras et al., 2017
*MAP2* (Mature neuronal marker)	Fw: GGAGCTGAGTGGCTTGTCAT Rv: CTAGCTCCAGACAGACGCAG	[21] Almeida et al., 2016
*SYP* (Synaptophysin; differentiated postmitotic neuronal cell marker)	Fw: TCCTCGTCAGCCGAATTCTTT Rv: CTCGCTACTTGTTCTGCAGGAA	[23] Jung et al., 2012
*Bcl-2* (Anti-apoptotic gene marker)	Fw: TCATGTGTGTGGAGAGCGTC Rv: TCAGTCATCCACAGGGCGAT	[21] Almeida et al., 2016
*Bax* (Apoptotic gene marker)	Fw: TCAGGATGCGTCCACCAAGAAG Rv: TGTGTCCACGGCGGCAATCATC	[24] Roh et al., 2022

## Data Availability

Not applicable.

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
