# Peer review of "An Optical and Chemiluminescence Assay for Assessing the Cytotoxicity of Balamuthia mandrillaris against Human Neurospheroids"

_bioengineering, 2022, doi:10.3390/bioengineering9070330_

Round 1

Reviewer 1 Report

The authors present a study regarding the an optical and chemiluminescence assay for assessing the cytotoxicity of Balamuthia mandrillaris against human neurospheroids. While the topic is of certain interest, this study presents some problems.

This study aims to demonstrate tumor spheroids as a better assay for evaluating the cytotoxicity of pathogenic amoebae, compared to conventionally two-dimensional cultured cells, however, the results presented do not corroborate this.

The study would benefit from further experiments, mainly aimed at improving statistical analysis.

The Discussion is difficult to follow in several places and should be revised.

Specific comments: 

Lines 68-69: …clinically isolated B. mandrillaris trophozoites at the pathogenic stage.Unclear meaning. If it is intended to indicate that a clinical isolate was used, specific information must be included in the materials and methods.

Lines 68-69: Regular culture of the clinical isolate of B. mandrillaris was performed following a previous study (Pengsart et al., 2022).The data in the references are missing, not available for consultation.

Line 139: B. mandrillaris…. it is not written in italics, to correct.

Line 159:  Figure 1: Scale bar, 100 mm. This data indicated in the caption is not associated with any bar in the figures.

In the graphs of Fig. 1, the significance is not expressed; in the caption, it is not clear whether the data refer to 3 replicates of a single experimental test or if they were obtained in different experimental tests. Statistical data must be added.

Line 195:  Figure 2: … ***p < 0.001;…  to be deleted, because it is not involved.

Line 208: … , the use of wortmanin … (correct is wortmannin); also indicate the concentration at which it was used.

Figure 3: In the graphs the significance is not expressed; in the caption, it is not clear whether the data refer to 3 replicates of a single experimental test or if they were obtained in independent experimental tests. In the case of replicates of a single experimental test, the standard deviation values seem quite high and from a graphical point of view they do not indicate clear significant differences. Therefore it is necessary to add the missing data.

Lines 222-225: Downregulation of the NEUROD1 gene was observed at 48 hours post-coculture. In contrast, the level of the proapoptotic BAX transcript was slightly decreased in the B. mandrillaris-cultured neurospheroid (Figure 3D). These results together suggest that B. mandrillaris likely affected cell viability. In the absence of statistical data indicating clear significant differences, it is incorrect to make these claims. 

Lines 272-275: Moreover, in the context of host-parasite interactions, the measurement of intracellular ATP in human neurospheroids is heavily confounded by the ATP of metabolically active trophozoites. Thus, a way to calculate the neurospheroid-specific ATP is proposed here. In this regard, I think it would be worth investigating further. It is known, for example, that free-living amoebas belonging to the genus Acanthamoeba release ATP (involved in cytopathogenicity mechanisms). It would be interesting to establish whether B. mandrillaris also exhibits the same ability. The kinetic evaluation of ATP production by trophozoites alone from 0 to 48h would certainly be useful to make the most of your results.

Author Response

Dear Reviewer,

We thank the reviewers for their generous comments on the revised manuscript and have edited the revised manuscript to address their concerns.

In particular, we have responded point-by-point to the reviewer’s comments below, with our responses immediately below the comment in blue color. To meet a standard format, we add more references. The revised portions are indicated in blue color. We believe that the revised manuscript is now suitable for publication in Bioengineering.

Yours sincerely,

Kasem Kulkeaw, Ph.D.

Reviewer 2 Report

This manuscript reports the development of a 3D assay for neuronal cytotoxicity assay for the rare pathogen Balamuthia mandrillaris. This assay will be useful for other similar human brain pathogens Naegleria and Acanthamoeba. The many advantages of the 3D systems are convincingly discussed.  The paper is well written and clear. A few minor points are given below.

The text in Figure 1 C is too small to read comfortably, also figure 3

Line 139 Italicise B. mandrillaris

Figure 4 The scale bar is too small to see

In the references Latin names should be italicised and also there are several instances where the genus name is not in capitals and other words such as Alzheimer’s (Line 393) do not have capitals?

Author Response

(The authors gave the same response as above.)

Reviewer 3 Report

In this study, the authors used human neurospheroids to mimic brain tissue and then cocultured with Balamuthia mandrillaris to explore the cytotoxicity They used tumor cell line to make spheroids, which is more convenient and cheaper than 2D/3D tissue models. However, there are some major issues need to fixed:

1. This study used spheroids formed by neuro tumor cells, how is this simple "cells cluster" model relevant to real human brain?  The authors didn't show any morphology of neuron cells in the spheroids, how did they know if the proliferation and differentiation is same as normal neuron cells in brain? I think the authors need to show the how their model replicate (or close to) the human brain structure.

2. The quality of confocal images are poor and it's very difficult to see the neuro cells or B. mandrillaris cells in the figure 4 and 5. Without clear images, it's impossible to tell how B. mandrillaris invade into spheroids. I suggest authors use high resolution images from more powerful lens (30x or higher) to show clearer edge of spheroids/cells.

3. This study used size-changing as a criteria for cytotoxicity, is there any relevant between this and actual pathogenesis of infection? I think Live/Dead staining plus confocal imaging should be a better method to assess the cells viability. 

Minor issues:

1. Line 26-35, please add references for introduction.

2. Line 45-47. please add references.

3. Line 48-50, please add references.

4. Line 101-102, the shape of spheroid is not perfect round or oval, how did author define "diameter"? Please add more detailed information for diameter measuring by ImageJ in methods section.

5. Line 150-151, how did author know that the dark zone was hypoxic core? Did they measure the oxygen concentration? Or have any reference?

6. Line 167-173, Spheroid expressed both immature and mature neuron. How did author think spheroid has more maturity? Also, add the p value in figure 1. Why did 2D result hasn't error bar? How many replicates did they do?

7. Line 190, Should be "without and with".

8. Line 222-225, how did author make the conclusion of viability changing based on only two genes' down-regulation since there were other 5 genes show less differences between coculture and control samples? And add the p-values in figure 3.

9. Figure 5, there were not labels of C and D.  

Author Response

(The authors gave the same response as above.)

Round 2

Reviewer 1 Report

The revised manuscript is undoubtedly more correct, clear and incisive.

Therefore, it deserves to be published.

Author Response

Dear Reviewer,

We thank for your generous comments on the revised manuscript. In the current version, the revised manuscript is suitable for publication in Bioengineering.

Yours sincerely,

Kasem Kulkeaw, Ph.D.

Reviewer 3 Report

The most of comments had been addressed. Could authors put their response to major comment 3 "There are several reports show necrotic lesion in an autopsied tissue of human brain parenchyma, suggesting a loss of neuron cells. Hence, the disappearance of the cells in the spheroid was used as end-point indicator of brain damage." into discussion part? Also please add the references. 

Author Response

Dear Reviewer,

We thanks for your suggestion. In the 2nd version of the revised manuscript, we added the sentences and the relevant references in the discussion (line 345-348). The revised portions are indicated in green color. We hope that the revised manuscript is now suitable for publication in Bioengineering.

Best regards,

Kasem Kulkeaw, PhD

"In the brain parenchyma, B. mandrillaris trophozoites must acquire nutrients from the surrounding neurons, leading to host cell damage. Together with attempts to restrict and remove the invading amoeba, host immune cells mount an inflammatory attack that collaterally damages the brain. An assay directly assessing neuronal damage allows findings of nonamoebicide targets to inhibit host cell damage. Thus, the mechanism underlying host cell damage is crucial for identifying a cell contact-dependent process in which the parasite must survive. Regarding 3D imaging, our previous work demonstrated the cellular process of host cell ingestion in a 2D culture of SH-SY5Y cells [20]. Given the difference in complexity of microarchitecture, the current study could not observe endocytosis of host cell protein by B. mandrillaris trophozoites in coculture with a 3D neurospheroid. With a limitation in live imaging of spheroids, the trophozoites first attached to the outer layer and then transformed into a round morphology lacking cytoplasmic projection. The 3D confocal images reveal the anchoring process, in which the cytoplasm of B. mandrillaris trophozoites protrudes into a layer of neurospheroid, while some cells are exposed to the culture medium. Moreover, some trophozoites completely submerged into the neurospheroid. Hence, searching for cell contact and invading factors may allow us to inhibit parasite survival. Given that several case studies reported the necrotic lesions in the autopsied tissues of human brain parenchyma [29] [30] [31,32], suggesting a loss of neuron cells. Therefore, the disappearance of the cells in the spheroid was used as end-point indicator of brain damage.